# Assessing the Usefulness of *Moringa oleifera* Leaf Extract as a Biostimulant to Supplement Synthetic Fertilizers: A Review

**DOI:** 10.3390/plants11172214

**Published:** 2022-08-26

**Authors:** Chuene Victor Mashamaite, Bonga Lewis Ngcobo, Alen Manyevere, Isa Bertling, Olaniyi Amos Fawole

**Affiliations:** 1Department of Agronomy, University of Fort Hare, Private Bag X1314, Alice 5700, South Africa; 2Postharvest Research Laboratory, Department of Botany and Plant Biotechnology, Faculty of Science, University of Johannesburg, P.O. Box 524, Johannesburg 2006, South Africa; 3School of Agricultural, Earth and Environmental Sciences, University of KwaZulu-Natal, Private Bag X01, Pietermaritzburg 3209, South Africa

**Keywords:** abiotic stress, biostimulants, chemical fertilizer, moringa leaf extract, plant growth regulation, sustainable agriculture, yield

## Abstract

The extensive use of synthetic chemical fertilizers is associated with environmental pollution and soil degradation. In addition, the high costs of these fertilizers necessitate the search for alternative, eco-friendly and safe natural sources of phytonutrients. The liquid extracted from moringa (*Moringa oleifera* Lam.) leaves has been used in agriculture to improve the growth and productivity of several crops. The efficacy of moringa leaf extract (MLE) is attributed to its high content of mineral nutrients, protein, vitamins, sugars, fiber, phenolics and free proline. In addition, MLE contains significant amounts of phytohormones, such as auxins, cytokinins and gibberellins. Furthermore, MLE is a valuable product promoting seed germination, plant growth and deeper root development, delaying fruit senescence and increasing the yield and quality of crops grown under normal or stressful conditions. Here, we review the research on MLE as a biostimulant to enhance crop growth and productivity. Moreover, we emphasize its possible introduction to smallholder farming systems to provide phytonutrients, and we further highlight research gaps in the existing knowledge regarding MLE application. Generally, MLE is an inexpensive, sustainable, eco-friendly and natural biostimulant that can be used to improve the growth and productivity attributes of various crops under non-stressful and stressful conditions.

## 1. Introduction

Fertilizers are substances used to improve crop growth and yield [1]. The dependency on synthetic chemical fertilizers for agricultural crop production is influenced by growing food demand, which is impacted by an exponentially increasing world population, economic development and climate change [2]. Consequently, chemical fertilizers have become a vital part of contemporary agriculture, providing the major plant nutrients, including nitrogen, phosphorus and potassium [3].

The excessive use of chemical fertilizers has been linked to soil degradation and environmental pollution [4,5,6]. The application of large quantities of chemical fertilizers has also been implicated in nutritional imbalances that encourage infestations of insects and crop diseases and stimulate the growth of problematic weeds [7]. Economically, intensive fertilizer use lessens the approval of crops for export [8]. Moreover, subsistence and smallholder farmers in various developing countries may not be able to afford the purchase of such fertilizers due to their costs [3]. Recently, the continuous search for alternative, reliable and safe natural sources of plant nutrients has gained interest in achieving sustainable agricultural practices while, at the same time, improving crop productivity [6,9].

As such, efforts to decrease the use of chemical fertilizer and increase nutrient use efficiency, as well as alleviate various biotic or abiotic stress on plants through the application of plant biostimulants, have been documented [1,5]. Plant biostimulants comprise numerous bioactive compounds that may improve diverse physiological processes, thus increasing plant growth and economic yield [10]. Nonetheless, the persistent usage of commercially available synthetic biostimulants is usually expensive (making them unaffordable to smallholder farmers) and less eco-friendly [11]. Alternatively, the combination of natural-product-based biostimulants with reduced levels of chemical fertilizers to accomplish optimum crop development has been suggested [12].

Recently, several new natural biostimulants have been used to improve the growth and productivity of a variety of crops. Moringa leaf extract (MLE), obtained from *Moringa oleifera* Lam. (colloquially known as moringa), is one such alternative crop, and is studied for its effect on the growth and yield of crops under standard and stressful conditions [4]. It naturally produces certain biostimulants, is easy to grow and has gained attention in the scientific literature [9,13]. Moringa is a plant in the Moringaceae family and is widely distributed across the tropical and subtropical regions, growing best in warmer growing environments [14]. With such a fast and perennial growth habit, moringa produces large biomass; its leaves are documented as being rich in mineral nutrients, protein, vitamins (A and C), amino acids, sugars, fiber, β-carotene, riboflavin, phenolics and free prolines [15,16,17]. These organs contain a significant amount of ascorbate and phytohormones, particularly the cytokinin zeatin, auxins and gibberellins [18,19]. Additionally, MLE exhibits high antioxidant activity, as it is a rich source of certain plant secondary metabolites, including osmoprotectants [20,21]. The balanced composition of phytohormones, antioxidants and mineral nutrients in moringa makes the extract an exceptional natural plant biostimulant [22,23]. 

The interest in growing MLE as a biostimulant for use in farming stretches from environmentalists to researchers and scientists worldwide [16]. There are various types of MLE, including aqueous extracts, pressurized hot water extraction and solvents. The ability of moringa, as an MLE, to contribute to the production of healthy and safe foods by means of environmentally friendly and sustainable agricultural practices plays a key role in assessing the nutritional benefits and market value of the leaf powder [24]. As such, enhancing the growth and yield of food crops using safe natural biostimulants, such as MLE, is imperative in the modern day [25]. Thus, its use can be promoted among farmers as a possible supplement or substitute for inorganic fertilizers. In this context, our review gathers information on MLE as a natural biostimulant to improve the growth, yield and quality of crops under standard and stressful growing conditions. We evaluated various methods of extracting bioactive compounds from moringa leaves, highlighting an economically friendly method that can be used by both commercial and smallholder farmers. Furthermore, the preservation methods of moringa leaves were assessed. Finally, we highlight the potential benefits of introducing it, particularly to smallholder farmers, as a means of phytonutrient application to improve plant growth and product quality while also identifying knowledge gaps that could be explored. To our knowledge, no review of this nature has been carried out on this important topic.

## 2. Brief Morphological Description and Uses of Moringa

Native to India, *M. oleifera* is the most widely distributed and utilized species worldwide compared to the other 12 moringa species making up the Moringaceae family [26]. Moringa is a small, drought-tolerant, fast-growing, and dicotyledonous tree that can reach a height of up to 4 m in a year and 6–15 m at maturity [14,27]. Cutting moringa trees frequently (i.e., every 40 days) result in a dry-leaf yield that ranges from 4.2 to 8.3 tons.ha^−1^ [28]. It is reported that about 578.24 billion tons of moringa leaf powder is sold annually worldwide, with Asia being the largest producer [29,30]. India has been identified as the main producer of moringa, with an area of 38,000 ha producing 1.1–1.3 million tons of tender fruits annually [31]. It is commonly known as the ‘drumstick tree’, ‘horseradish tree’, ‘mother’s best friend’ or ‘miracle tree’ in English and ‘*Makgonatšohle*’ (in Sepedi, meaning ‘the ability to do everything’) [15,32]. It has multiple uses, including in human nutrition, human and veterinary medicine, livestock feed, cosmetics and water purification, and as a bio-fertilizer, biostimulant, antioxidant and antimicrobial agent [23]. 

## 3. Brief Overview of Plant Growth Regulators and MLE as a Natural Biostimulant

Plant hormones are defined as ‘small chemicals that play a crucial role in plant growth and development’ [33]. Plants produce various hormones, namely: abscisic acid, auxins, brassinosteroids, cytokinins, ethylene, gibberellins, jasmonic acid, salicylic acid and strigolactones [33,34]. Plant growth regulators (PGRs) are applied to plants to moderate their growth and increase their nutritional efficiency, quality, stress tolerance and rhizospheric activities [10]. The exogenous application of PGRs has been effectively used to promote or retard crop growth and foster plant tolerance to biotic and abiotic stress [10,35]. These functions include flower induction and hastening maturity or senescence [36]. Synthetic PGRs are, however, often expensive, and very slowly biodegraded [24]. Due to the eco-toxicological effects of these synthetic compounds [37], the application of natural plant biostimulants in agriculture has increased [13,38,39]. 

Natural biostimulants have been gaining interest in sustainable agriculture to enhance crop productivity and quality [40]. Biostimulants comprise a wide range of compounds capable of increasing certain physiological processes which encourage crop growth and development, with the intention of reducing the use of synthetic fertilizers without negatively affecting crop productivity [12,13,41]. The most utilized natural biostimulants in crop production are biochar, chitin, chitosan derivatives, fulvic acid, humic acids, microbial inoculants, MLE, plant extracts, protein hydrolysates and seaweed extracts [10,42]. Many of these biostimulants are costly to use on a commercial scale of production [5].

Among these natural products, MLE has received wide recognition in the scientific community as an alternative, safe source of biostimulants, particularly due to an abundance of growth-promoting substances [22,43]. Moringa leaves contain a wide range of vitamins, essential minerals and amino acids (Table 1 and Table 2). Other moringa plant parts also contain considerable amounts of phytochemicals, but their use in plant production is not as popular as the use of moringa leaves. Several studies have evaluated the efficacy of MLE on the growth, yield and quality of crops [44,45,46,47], but only a few have analyzed the hormonal profile of the moringa extracts (Table 3). In addition, the concentrations of the main hormone in MLE, zeatin, were found to be a thousand times (from 5–200 μg·g^−1^) higher than the zeatin concentrations in other plant species [16,27,48]. According to Nasir et al. [39], the effectiveness and efficacy of MLE as a biostimulant are equivalent to or slightly higher than in synthetic PGRs. Moreover, the extract is easy to prepare, less costly, and environmentally friendly [49]; thus, it is a crop stimulator that subsistence farmers can use.

As mentioned earlier, the ability of MLEs to enhance the growth and yield of various crops is attributed to the bioactive compounds present in their extracts (Table 3). These compounds or active ingredients can enhance certain key physiological, biochemical and molecular processes. These mechanisms increase the availability of nutrients in the soil or plant and positively affect the growth and yield parameters, as well as the nutritional quality of leaves, pods or fruits. The application of MLEs also enhances photosynthesis and further improves the metabolism of carbon and nitrogen [57,58]. Furthermore, the presence of phytohormones in the MLEs plays a crucial role in cell division, resulting in cell multiplication and general cell enlargement or elongation, ultimately leading to the improved growth and yield of crops [25,59].

## 4. Methods and Preparation of MLE

Recently, the use of moringa plant parts, particularly its leaves, has gained popularity in various sectors as a dietary supplement and herbal medicine, as well as being used as a growth stimulant in plant production. The concentration and timing of MLE application differ with the scope of the study [27,46,47], and the response of the crop differs with the type of MLE extract, the extraction solvent used and the concentration of the extract [39,46,47,48,56]. It is, therefore, important to review the extraction methods and solvents used to determine how varying the extraction process can result in the leaf extract containing high or low concentrations of ‘active ingredients’ (biological compounds) that can enhance either nutraceutical or agricultural food production.

### 4.1. Preparation of Aqueous Moringa Extracts

Berkovich et al. [60] extracted phytochemical compounds and minerals from the dry leaf powder of moringa trees grown in tropical and subtropical areas with rich mineral soil. The derived aqueous moringa extract was prepared by mixing 1 g of dry moringa leaf powder with 10 mL boiling water for 5 min before filtering it twice through 2 μm sterile filter paper into a sterile tube. This aqueous extract stock solution (100 mg/mL) was freshly prepared for each set of experiments and stored for no longer than 5 days at 4 °C [60]. Instead of using dry leaf powder, Elzaawely et al. [25] collected fresh moringa leaves from trees grown at Tanta University’s experimental farm, Egypt. Their study adopted methods by Phiri and Mbewe [48], who prepared the aqueous extract of moringa at a 1:10 (*w*/*v*) ratio and mixed 30 g of fresh leaves (instantly after collection) with 300 mL of distilled water in a household blender for 15 min. Subsequently, the authors filtered the obtained solution through muslin cloth and diluted it with distilled water at distinct ratios (1:20, 1:30, and 1:40) immediately prior to application onto the crops. Although water as a solvent does not result in a high yield, generally, the aqueous extraction of moringa has gained popularity in recent years amongst farmers, especially subsistence and smallholder farmers, because water is most readily available, environmentally friendly and cheap compared to other solvents such as methanol and alcohol [57,61,62,63]. 

### 4.2. Pressurized Hot Water Extraction (PHWE) of Moringa Leaves

This extraction method, not using any organic solvents, was designed by Nuapia et al. [64]. These authors used an extraction system using hot water pressure as described by Matshedido et al. [65] with some modifications. Briefly, in each run, a PHWE cell was filled with a mixture of 5 g of moringa leaf powder and 5 g of diatomaceous earth. A PHWE cell loaded with the moringa leaf powder and diatomaceous earth was pre-heated for 10 min. During the extraction process, the extraction conditions differed in terms of temperature and time. Aluminum foil was then used to cover the vessels collected during the extraction process, and these vessels were kept at a low temperature (in a cooler box filled with ice) to avoid the degradation of any sensitive bioactive compounds. The moringa leaf extracts were then collected after each extraction cycle, kept in the centrifuge tubes, and stored at 18 °C for further analysis of bioactive compounds. It must be noted that even though water is the most naturally occurring liquid, its use in this process of extraction causes difficulty in concentrating the extracts. This is due to the high heat of vaporization of water compared to organic solvents such as methanol and ethanol. In addition, in rare cases where water solvent needs to be removed via evaporation, the extraction process requires a lot of energy [66].

### 4.3. Preparation of MLE Using Chemical-Based Solvents

Ngcobo and Bertling [47] prepared MLE according to Makkar and Becker [67], with slight modifications. Concisely, 20 g of fresh, young moringa leaves were mixed with 675 mL of 80% methanol. The suspension was homogenized to maximize the amount and types of phytochemicals extracted before the solution was then filtered through Whatman No. 2 filter paper. The extract was diluted with distilled water to three concentrations (20, 50 and 80% of the originally obtained MLE). These dilutions were sprayed directly onto the leaves of cherry tomato plants to run off [47]. In contrast, Vongsak et al. [68] crushed both dried (1:20, *w*/*v*) and fresh (1:20, *w*/*v*) moringa leaves into small pieces and soaked them in 70% ethanol for 72 h at room temperature (28 ± 2 °C) with occasional shaking. The extract was then filtered, and the residue was re-soaked with the same solvent amount until the extraction was clear. Organic solvents are the most commonly used method of extracting compounds from moringa plant parts, with many authors adopting this approach [44,45,53,61]. Furthermore, the chemical-based solvents that are commonly used to extract the bioactive compounds in moringa leaf are volatile, leading to a certain percentage of the foliarly applied MLE not being taken up by plants due to evaporation. To overcome this, soil drenching or the application of MLEs before sunrise or after sunset is recommended. Moreover, the dilution of chemical-based extracts to a lower percentage is recommended since methanol, in particular, is highly toxic.

## 5. Preservation Methods of Moringa Leaves

Moringa leaves can be stored and used throughout the year without losing their concentration of phytochemical nutrient quality [69]. Storing the moringa plant parts, including the leaves, allows them to be available even for off-season uses [70]. The oldest methods of preserving the leaves include either drying or freezing. Drying is the most effective method of preserving moringa leaves, as it reduces the moisture content to a percentage that prevents enzymatic reactions and non-enzymatic oxidation quickly [71]. Therefore, it is pivotal to choose an eco-friendly drying method to remove moisture to maintain the biological activity of the leaves. Additionally, the method employed to dry moringa leaves must retain the original, high phytonutrients at a level comparable to that of fresh leaves. An investigation by Foline et al. [72] compared various methods of drying (multipurpose dryer, (50 °C), air drying (room temperature) and sun-drying). The study revealed that using this multipurpose dryer may be the best method of drying moringa leaves, as the extracts from such leaves retained most of their phytonutrients, and only air-drying maintained higher levels of such compounds.

Saini et al. [73] compared drying methods of fresh moringa leaves, namely cabinet tray-, oven-, sun- and microwave-drying, and lyophilization. Their findings suggested that cabinet tray-drying was as efficient as lyophilization in retaining the highest concentration of total carotenoids (60.1%), especially *trans*-*β*-carotene (90.1%) and 13-*cis*-lutein (93.2%), as well as DPPH (2,2-diphenyl-1-picryl-hydrazyl-hydrate) activity; however, *trans*-lutein (51.3%) and ascorbic acid (97.8%) were best preserved in the leaves via lyophilization. Furthermore, oven-drying instead of sun-drying was recommended at the household level to preserve nutrients and antioxidant activity in moringa leaves [73]. Various methods of drying moringa leaves were compared by Yang et al. [74], who revealed that drying moringa leaves after blanching using a heat-pump-driven dehumidifier resulted in relatively greater leaf powder quality compared with hot-air tray drying and microwave drying. The same authors further revealed that the retention of moringa nutrients, except for vitamin C, was achieved using low-temperature oven-dried leaves. However, the formulation of MLE as a marketable, commercial product will require further studies on effective preservation methods to guarantee the stability of the secondary metabolites found in moringa leaves [75].

## 6. Moringa Leaf Extract as a Plant Growth Enhancer 

MLE has been identified as one of the most significant natural biostimulants due to its abundant minerals, phytohormones and antioxidants [9,17]. The preparation of this product using water [60] or 80% methanol [47] is both eco-friendly and cost-efficient, even though some farmers are still concerned about the toxicity of methanol. The extract of moringa leaves has been used as a plant growth enhancer on several crops, including wheat, citrus and tomato [13,39,47]. Moringa leaf extract is considered a valuable product for promoting seed germination, vigorous plant growth and deeper root development, while also delaying fruit senescence and increasing yield and product quality [19]. These improvements are ascribed to moringa leaves being rich in phytochemicals. Other studies on MLE as a growth stimulator described yield and nutrient uptake enhancements in various agronomic and horticultural crops [34,36]. It is, however, important to note that different plant species react differently to various MLE concentrations [76]. The responses of various crops to MLE application are summarized in Table 4 below, followed by a detailed discussion on how different crops respond to the application of MLE.

### 6.1. Effect of MLE on Studied Cereals

#### 6.1.1. Effect of MLE on Growth, Yield and Quality of Wheat

The high cost of and unreliable accessibility to inorganic fertilizers have affected the production of cereal crops [4], particularly for smallholder farmers in rural areas. Wheat (*Triticum aestivum* L.) is the third largest cereal crop, after maize and rice, being produced worldwide for human nutrition and industrial uses [77,85]. Due to this importance, increasing wheat production using suitable management practices is of high importance [51]. The foliar application of MLE to wheat positively influenced plant height, fresh and dry root mass and certain yield components [9,52]. Merwad and Abdel-Fattah [51] showed that MLE treatments (derived from 20 g of moringa leaves mixed with 675 mL of 80% ethanol) significantly enhanced wheat yield parameters, such as total biomass, 1000-grain weight, straw and grain yield and yield efficiency, as well as protein content compared with control plants. Moreover, the highest values of these parameters were observed using a 4% MLE application under field conditions. Similarly, the application of MLE (stock solution of 10 mL water with 320 g of fresh moringa leaves) during critical growth stages of late-sown wheat (i.e., in the tillering, jointing and booting stages) produced the highest grain and straw yields under field conditions [9]. The foliar application of 3% aqueous MLE was more effective in enhancing the growth, the biological and grain yields and the biochemical parameters of wheat than the control [75,77]. This was ascribed to the presence of ascorbic acid and phenolics in the MLE [75]. According to Khan et al. [75,77], the use of fresh aqueous MLE at 3% as a seed-priming agent and foliar application at the tillering and boot stages could help in attaining uniform stand establishment, as well as higher biological and economic yields of normal and/or late-sown wheat. 

#### 6.1.2. Moringa Leaf Extract on Maize and Sorghum Crops

Maize (*Zea mays* L.) and sorghum (*Sorghum bicolor* L.) are important grain crops for meet the high global food demand and livestock feed needs [86,87]. As such, it is important to improve their production using sustainable practices. The effectiveness of MLE as the foliar fertilizer to improve growth, as well as grain and stover yields, has been established in maize field trials [34]. Other authors have also demonstrated that when MLE was foliarly applied two weeks after emergence, as well as two weeks later, there were significant increases in growth parameters such as plant height, fresh and dry shoot mass, number of grains, 100-grain mass and grain mass per plant [78]. Bashir et al. [79] showed that the growth and yield of sorghum could be increased by the application of MLE. The contact of MLE with cereal seeds improved the germination of sorghum (by 29%) and the length of maize radicals [4]. The response of crops to MLE treatment does, however, depend on the frequency of application and the crop species. In some crops, a higher frequency of application results in a better yield [9,34]. As such, Biswas et al. [78] suggested that a more frequent application of MLE to maize plants would result in better plant growth and improved yield. Generally, grain yield and the quality of cereal crops are critical factors, guaranteeing successful economic production and marketing, as high-quality standards are set out by the market [88].

### 6.2. Effect of MLE on Legumes 

Food legumes play a vital role in the agricultural sectors, particularly in developing countries, because of their ability to produce substantial amounts of protein-rich seeds for human nutrition and well-being [18]. Moringa leaf extract improved fruit size and sugar concentration when foliarly sprayed onto soybeans (*Glycine max* L.) [27]. Likewise, a study by Mvumi et al. [34] showed that MLE increased the growth and yield of common beans (*Phaseolus vulgaris* L.) under both greenhouse and field conditions. The effect of MLE on the germination and seedling survival of three common legumes, namely bean, cowpea (*Vigna unguiculata* (L.) Walp.) and groundnut (*Arachis hypogaea* L.), was also examined by Phiri and Mbewe [48]. The outcomes of this study indicated that the application of MLE enhanced the germination of beans and cowpeas and promoted the survival of all three legume seedlings. 

Elzaawely et al. [25] showed that the vegetative growth, biochemical attributes, yield and quality of snap beans (*Phaseolus vulgaris* L.) were enhanced when sprayed with MLE. According to these authors, the effect of MLE on crops is dose dependent. This is supported by Merwad’s [45] findings when investigating the effect of levels of MLE (0, 1, 2, 3 and 4%) on pea plants; they showed that the highest values of photosynthetic pigments, dry shoot mass, biological yield, pod yield and nutrient accumulation were obtained following the application of 4% MLE. The extract solution was prepared by mixing 20 g of fresh leaves with 675 mL of 80% ethanol. Elzaawely et al. [25] found that the dilution of aqueous MLE, prepared by mixing 30 g of fresh leaves with 600 mL distilled water using a household blender for 15 min (1:20), resulted in better growth and yield in snap beans than other treatments. The phytohormonal concentrations of snap bean leaves (i.e., auxins, cytokinins, gibberellins, jasmonic acid and salicylates) were markedly increased in response to the foliar application of diverse MLE concentrations. The improvement in growth and yield parameters and the productivity of snap beans following MLE application were attributed to high levels of these phytohormones [25]. 

### 6.3. Effect of MLE on Tomato

Tomato (*Solanum lycopersicum* L.) is the most-produced horticultural commodity in the world, and its demand is continuously increasing due to the health benefits associated with its consumption [89]. Several studies explored the potential benefits of MLE on improving tomato productivity. This includes studies by Culver et al. [44], Mvumi et al. [80,90], Ngcobo and Bertling [47] and Hoque et al. [81], which found that MLE significantly increased the plant height, leaf number, number of branches, above-ground dry biomass, dry root mass, fruit yield and quality of tomato. The extracts from fresh moringa leaves influenced tomato productivity by augmenting its leaf lamina thickness, stomatal density, stomatal size and fruit yield [80]. Additionally, MLE application improved the productivity of tomato leaves infected with pathogenic organisms (i.e., *Alternaria solani*), as depicted by the stomatal density showing a strong correlation with the fruit yield [80]. The hormone cytokinin found in MLE affects tomato plant transpiration by regulating stomatal density and the size of the leaves [91]. The magnitude of the impact MLE application has on crops is proportional to the concentration used [83]. Generally, the application of MLE from two weeks after transplanting up to physiological maturity is recommended for improving tomato plant productivity in greenhouse and field crops [80,90].

### 6.4. Effect of MLE on Citrus

Citrus is one of the major horticultural crops grown worldwide, with 144 million tonnes being produced in 2020 [92]. Nonetheless, major challenges encountered by producers of “Kinnow” mandarin (*Citrus nobilis* Lour × *C. deliciosa* Tenora) are its low productivity and its poor fruit quality. In attempts to unravel these challenges, Nasir et al. [39] assessed the effect of various MLE concentrations (2, 3, 4 and 5%) on “Kinnow” mandarin trees. It was revealed that the application of 3% (3 mg/mL) MLE at different growth stages effectively reduced fruit drop and successively increased fruit set, fruit size, fruit yield and certain biochemical quality characteristics [39]. In another study, the foliar application of 3% MLE, together with 0.25% K_2_SO_4_ and 0.6% ZnSO_4_, at the fruit set stage also increased fruit yield components, as well as leaf macro- and micronutrients (N, P, K, Ca, Mn and Zn) and the ascorbic acid concentration in “Kinnow” mandarin leaves [19]. In contrast, 3% MLE foliar application alone increased mineral and ascorbic acid levels in “Kinnow” mandarin leaves [39]. The enhancement of “Kinnow” mandarin leaf nutrient concentrations after the foliar application of MLE could be ascribed to MLE being a rich source of mineral nutrients [93]. Thus, MLE could enhance endogenous mineral levels through simple diffusion or stomatal conductance. To increase or maintain citrus tree performance, trees should be supplied with nutrients in a balanced manner, along with growth promoters such as MLE, to produce high-quality fruit [94].

### 6.5. Effect of MLE on Other Plant Species

Ndubuaku et al. [82] reported that growth parameters such as plant height and leaf number in cassava plants (*Manihot utilissima* Pohl.) were considerably increased following foliar MLE application. According to Hoque et al. [81], applying MLE can bring immense benefits to the crop produced. Application of aqueous MLE at 100 mL crude extract (100 mL distilled water/1000 g fresh leaves) diluted with 200 mL distilled water resulted in the highest germination percentage, seedling emergence, seedling survival, plant height, dry shoot mass, dry root mass and yield components in sunflower plants (*Helianthus annuus* L.) [46]. Indian spinach (*Basella alba* cv. Red Malabar) showed enhanced growth and a 20% increase in yield after the application of aqueous MLE (stock solution = 10 g fresh leaves with 1 mL distilled water) at 25 mL/plant [81].

Hegazi et al. [83] revealed that the foliar application of aqueous MLE at 6 g/L produced better growth and seed yield in squash plants (*Cucurbita pepo* L.). Moringa leaf extract, applied at rates of 2% or 3%, encouraged the biomass of rocket plants (*Eruca vesicaria* subsp. *sativa*). It also enhanced the leaves’ photosynthetic pigments, total protein, total sugar, phytohormones (auxins, cytokinins and gibberellins) and several essential mineral nutrients, such as N, P, K, Ca, Fe and Mg [5]. Abou El-Nour and Ewais [84] demonstrated that applying MLE solution (1 kg moringa leaves mixed with 1 L of 80% ethanol) at 4% (40 mL crude extract with 960 mL distilled water) as a soaking treatment for 6 h to pepper (*Capsicum annuum* L.) seeds stimulated the germination percentage and rate and the germination index, as well as the germination velocity. The same MLE concentration, applied as a foliar spray, also improved the seedling growth of pepper crops in the nursery and produced strong plant growth, a high fruit yield and a high fruit nutrient concentration in open field peppers [84].

The foliar application of aqueous MLE at 5% (mixture of 100 g of fresh leaves with 1 L of distilled water) gave the best vegetative growth and yield characteristics in fennel (*Foeniculum vulgare* Mill.) [24]. When applied to geranium (*Pelargonium graveolens* L. Herit), various concentrations of aqueous MLE solution (diluted from 10 kg fresh material per 1 L water) significantly increased the plant height, branch number, leaf area, overall plant biomass and volatile oil content, as well as its constituents, geraniol and citronellol [6]. In addition, the photosynthetic pigments (total chlorophyll and carotenoids), phenolics, radical scavenging activity and nutrients in geranium leaves were enhanced after the application of 1:20 (10 g·mL^−1^ crude extract diluted with 20 mL water) MLE treatment [6]. Ethanolic extracts of moringa (20 g of dried leaves with 675 mL of 80% ethanol) diluted with distilled water at 5 g/L exhibited similar plant growth promoter attributes on the growth and yield of basil plants (*Ocimum basilicum*) [54]. Other plant species that were positively influenced by the application of MLE include, but are not limited to, coffee (*Coffea arabica* L.), sugarcane (*Saccharum officinarum* L.), cantaloupe melon (*Cucumis melo* var. sanctum) and blackgram (*Vigna mungo* L. Hepper) [27,67].

Among the phytohormones detected in MLE, cytokinins (zeatin) play an important role in promoting cell division and modifying apical dominance [37]. The diluted extract at different concentrations also stimulates shoot initiation and growth, chlorophyll synthesis and nutrient uptake, and delays senescence processes and aging in numerous plant tissues of various crops [59] such as cucumber (*Cucumis sativus* L.) [95]. Gibberellins, on the other hand, increase plant height and photosynthetic efficiency, cause broader and elongated leaves and enhance biomass accumulation [96]. Auxins found in MLE promote cell elongation, stem growth, lateral root formation and fruit development in crops [37]. Therefore, MLE may be foliarly applied as an environmentally friendly biostimulant to enhance the growth and productivity of several crops. 

## 7. Potential of MLE to Induce Plant Tolerance to Abiotic Stress

Besides its benefit of improving crop productivity under normal growing conditions, MLE has also been used to mitigate the negative effects of environmental stress [75]. Several studies have shown the potential of MLE in improving crop resistance and/or tolerance to abiotic stressors, such as drought, high temperatures, chilling temperatures, heavy metals and salinity [16,18,22,52,57].

### 7.1. Heat and Low-Temperature Stress

Global warming is one of the most alarming issues worldwide, and it is anticipated that rising temperatures will have adverse effects on agricultural crop production and, consequently, on food security [97]. Schmidhuber and Tubiello [98] projected that increasing heat stress will affect crop yield, and this will result in higher risks of hunger by 2080. In efforts to reduce the impact of these risks associated with climate change on crop quantity and quality, Afzal et al. [99] recently investigated the use of MLE to mitigate the impact of heat stress on late-planted wheat. This field study was conducted because late-planted wheat is frequently exposed to high temperatures during grain filling, resulting in the deterioration of overall crop performance. The authors showed that the exogenous application of 3% aqueous MLE increased the heat tolerance of wheat crops by promoting growth, development, secondary metabolite accumulation and relative water content and reducing oxidative damage [99].

Contrary to heat stress, low temperature is also one of the most important plant abiotic stressors which inhibit plant growth and productivity [100]. Chilling temperatures can cause several metabolic and physiological disturbances in the plant cells of warm-season crops, resulting in chilling injury and the death of certain plant seedlings [21]. Due to their tropical and subtropical origin, the seedlings of moringa are highly susceptible to low temperatures [21]. As such, Batool et al. [21] showed the potential for 3% aqueous MLE to improve the growth and yield of moringa seedlings under low temperatures. The MLE application enhanced the number of branches by 92%, leaves by 141% and leaflets by 61%, the phenolic concentration by 78%, and the overall yield compared with untreated moringa plants. This was attributed to the high amounts of antioxidants, growth hormones and mineral nutrients found in MLE.

### 7.2. Drought Stress

The negative effects of drought stress on crops can be reduced by applying natural biostimulants, such as MLE [35]. For example, the dry shoot and root mass, root length, leaf soluble proteins, leaf water content and grain yield of maize were significantly increased after MLE treatment using 25 mL distilled water with 250 g of fresh leaves per plant under drought stress [81]. Furthermore, the accumulation of cell-wall-bound phenolics caused by drought stress was drastically reduced under MLE application [81]. Similarly, aqueous MLE (stock solution: 10 kg leaves/L distilled water) applied at 3% was effective in mitigating drought stress damage to squash plants through the retention of a higher relative water content, higher osmoprotectant concentrations and a higher water-use efficiency, as well as displaying less electrolyte leakage [41]. When applied exogenously to drought-stressed plants, MLE alters the plant phenotypic response and metabolic processes that improve plant growth and productivity [57,101]. Thus, MLE effectively alleviates drought stress, seemingly by altering the anatomical and physiological characteristics of plants.

### 7.3. Salinity and Heavy-Metal Stress

Previous studies have verified the potential of MLE to improve the resistance and tolerances of various crops to stress associated with salinity and heavy metals (such as cadmium) [18,102]. These studies include experiments conducted by Yap et al. [53], which demonstrate that foliar MLE application at 10 g/L reduces salinity-induced adverse effects on the growth, yield and silybin content of *Silybum marianum* (L.) Gaertn. (milk thistle plants) irrigated with a 4000 ppm saline solution. In another inquiry, aqueous MLE-treated squash plants were found to have a higher relative water content, more antioxidants and lower electrolyte leakage; these features play a vital role in plants’ development, metabolism and responses to salinity stress [41]. According to Howladar [18], MLE-treated bean plants showed a significant increase in their level of antioxidants (such as carotenoids) and antioxidant enzymes (including catalase, glutathione reductase, peroxidase and superoxide dismutase), as well as proline, under both salinity and high-cadmium conditions. In addition, Khalofah et al. [103] showed that the application of aqueous MLE increased enzymatic and non-enzymatic oxidation and antioxidant activity in *Lepidium sativum* Linn. (Garden cress) plants under cadmium stress. Cadmium poses a great threat to any ecosystem, as it contaminates both plants and soils [18], and application of MLE could help to mitigate the adverse effects of this heavy metal. The application of MLE increased the tolerance of cadmium-stressed plants by enhancing their enzymatic and non-enzymatic antioxidant systems [104], thus improving growth and productivity under stressful conditions. 

## 8. Potential Benefits of Introducing MLE to Smallholder Farmers

The production of healthy and safe food using environmentally friendly and sustainable agricultural practices, such as MLE application, plays an essential role in assessing their nutritional and market value [24]. As such, efforts should be made to introduce the application of MLE to small-scale (and even commercial level) farmers. Smallholder crop production in many countries experiences challenges due to the insufficient use of mineral fertilizers and low inherent soil fertility due to nutrient-depleted soils [105]. These factors limit soil productivity on arable farmlands, and most farmers are unable to afford agricultural inputs, such as chemical fertilizers, due to their costs [86]. Furthermore, the continuous use of high amounts of chemical fertilizer has been documented to be harmful to the environment, potentially resulting in increased soil acidity, nutrient imbalances and reduced crop yield and quality [106]. Extracts from moringa leaves could provide an effective, available and cheap source of natural biostimulants for smallholder farmers and should even be included in commercial farmers’ crop production programs, thus promoting a green economy.

## 9. Considerations, Knowledge Gaps and Recommendations

The chemical composition of moringa leaves is influenced by various factors, such as the plant’s genotype, part, growth stage, growing conditions and postharvest treatments [107]. It is also important to note that biostimulants such as MLE are not fertilizers meant to correct a severe nutrient deficiency, as they do not comprise supplements intended to be directly applied to plants [108]. Biostimulants are, however, substances that, in small amounts, encourage mineral procurement and promote plant growth through their direct effects on metabolic processes [109,110]. The major application of these products is directed toward improving plant growth while also assisting plants during abiotic stress, to improve plant resilience, and, subsequently, enhance the yield and quality of the produced commodity [108]. Moringa plant parts, particularly the leaves, contain growth-promoting substances such as ascorbic acid, phenolics, natural antioxidants, various mineral nutrients and growth hormones; these bioactive compounds make MLE an ideal plant growth promoter. It must, however, be noted that there are various extraction methods and solvents used to extract the active ingredients in moringa leaves, and aqueous extraction is recommended as it is environmentally friendly and can be used by both subsistence and commercial farmers, as well as smallholder farmers in marginal communities. As such, the application of MLE should not be viewed in isolation but should be observed together with other crop management practices that effectively increase plant growth [110,111]. This view is important since MLE is not intended for use as a fertilizer but as a biostimulant. 

With regard to MLE as a plant protectant against abiotic stress, further studies are necessary to assess the accuracy of its functions in signaling pathways and in physiological responses to abiotic stress. In addition, the effect that MLE application has on soil micro-organisms has not yet been investigated. Soil micro-organisms play a significant role in agriculture, animal and plant health, the global food web and carbon and nutrient cycling [112]. It is very important to assess the effect of MLE on this web, as it has been observed that the use of synthetic fertilizer can have detrimental effects on soil microbial-species richness and distribution [113]. Because of their significant role in the ecosystems [114], the potentially beneficial or adverse effects of MLE application on these species should be determined. Finally, to avoid any bias, the use of MLE as a plant growth stimulator should not be limited to agronomic and horticultural food crops but should also be tested on medicinal and aromatic plants. Instead of applying moringa biostimulants in the form of MLE, it would also be interesting to evaluate the response of agricultural crops to moringa leaf powder (MLP), applied individually or incorporated into synthetic fertilizers and applied directly to or incorporated into the soil.

## 10. Conclusions

Recently, MLE has received enormous attention as a biostimulant due to its high concentration in phytohormones, antioxidants, macro- and micro-mineral nutrients, amino acids and osmoprotectants. This review summarized the development of research on its preparation, use and applications as a biostimulant. Based on scientific reports, MLE is an inexpensive, easily accessible, sustainable and eco-friendly natural biostimulant that can be used to improve the growth and productivity attributes of various crops under non-stressful and stressful conditions. Generally, the extract of moringa leaves has the potential to be used as a supplement to inorganic fertilizers and, therefore, should be promoted for application by small-scale and commercial farmers.

## Figures and Tables

**Table 1 plants-11-02214-t001:** Nutrient composition of fresh moringa leaves.

Component	Value (mg·100 g^−1^)	Reference
Calcium	440–2800	[6,50,51,52]
Potassium	259–2510	[6,50,52]
Magnesium	42.0–670	[23,51,52]
Phosphorus	70.0–390	[6,51]
Copper	4.00–14.0	[6,52]
Zinc	5.00–27.0	[6,53]
Nitrogen	1240	[6]
Sulfur	137	[50]
Manganese	84.0–396	[6,52]
Sodium	75.0	[52]
Iron	37.0–160	[6,23,52]
Vitamin A (β-carotene)	6.78–20.0	[51,53]
Vitamin B1 (Thiamine)	0.21–2.60	[50,53]
Vitamin B2 (Riboflavin)	0.05–21.0	[50,51]
Vitamin B3 (Nicotinic acid)	0.80	[53]
Vitamin C (Ascorbic acid)	220–847	[6,50]
Vitamin E (Tocopherol acetate)	77.0–448	[23,45,51]
Protein	6700–27,300	[50,51]
Fiber	900	[50]
Carbohydrate	12,500	[23]
Fat	1700	[23]

**Table 2 plants-11-02214-t002:** The nutrient composition of dried moringa leaf powder.

Component	Value (mg·100 g^−1^)	Reference
Calcium	1721–2185	[23,32,50]
Potassium	1324–2770	[32,41,45,50]
Magnesium	334–448	[23,32]
Phosphorus	70.0–550	[23,32,50,54]
Copper	0.49–21.0	[23,41,53,55]
Zinc	3.10–45.0	[41,55]
Iron	25.6–189	[23,41,45,50]
Nitrogen	31.3	[41]
Sulfur	268–870	[41,50]
Manganese	4.90–97.0	[41,56]
Sodium	31.0	[32]
Vitamin A (β-carotene)	6.78–16.3	[45,54,56]
Vitamin B1 (Thiamine)	2.02–2.64	[5,23]
Vitamin B2 (Riboflavin)	20.5–21.3	[5,23,56]
Vitamin B3 (Nicotinic acid)	7.60–8.20	[23,56]
Vitamin C (Ascorbic acid)	15.8–220	[5,23,45,54]
Vitamin E (Tocopherol acetate)	113	[5,45,56]
Protein	27 100–29 400	[45,50]
Fiber	12 500–19 200	[23,50]
Carbohydrate	12 500–41 200	[23,54]
Fat	1700–5300	[23,54]

**Table 3 plants-11-02214-t003:** Phytohormonal profile and osmoprotectant concentrations of moringa leaves.

	Component	Value	Reference
**Fresh Leaves**	**Phytohormones (μg·g^−1^)**		
	Indole acetic acid	0.44–0.83	[6,25,52]
	Zeatin	5.00–200	[16,27,54]
	Gibberellins	0.65–0.74	[6,52,53]
	Abscisic acid	0.13–0.29	[6,52]
	Salicylic acid	1.87	[6]
	Trans-jasmonic acid	0.22	[25]
	**Osmoprotectants (mg·g^−1^)**		
	Total amino acid	106–388	[6,52]
	Proline	21.0–33.7	[6,52]
	Total soluble sugars	249–352	[6,52]
**Dried leaves**	**Phytohormones (mg·g^−1^)**		
	Indole acetic acid	0.83	[52]
	Zeatin	0.03–0.96	[52,54]
	Gibberellins	0.003–0.054	[41,54]
	Abscisic acid	0.29	[52]
	Salicylic acid	0.082	[41]
	**Osmoprotectants (mg·g^−1^)**		
	Total amino acids	300	[41]
	Proline	30.0	[41]
	Total soluble sugars	170	[41]

**Table 4 plants-11-02214-t004:** Effect of applying moringa leaf extract to various crops.

Crop	Key Findings of the MLE Effect	References
*Triticum aestivum* L.	Improved plant height, fresh and dry root mass, above-ground biomass, 1000-grain weight and straw and grain yield	[9,52,75,77]
*Zea mays* L.	Increased plant height, fresh and dry shoot mass, number of grains, 100-grain mass and grain mass/plant	[34,78]
*Sorghum bicolor* L.	Improved germination, plant height, biomass and grain yield	[4,79]
*Glycine max* L.	Increased fruit size and sugar concentration	[27]
*Phaseolus vulgaris* L.	Increased vegetative growth, photosynthetic pigments, dry shoot mass, pod yield and phytohormonal concentrations of leaves	[25,34]
*Vigna unguiculata* (L) Walp.	Improved germination and seedling survival	[48]
*Arachis hypogaea* L.	Improved germination and seedling survival	[48]
*Solanum lycopersicum* L.	Increased plant height, leaf number, number of branches, dry shoot biomass, dry root mass, leaf lamina thickness, stomatal density, stomatal size, fruit yield and fruit quality	[47,80,81]
*Citrus nobilis* Lour *× C. deliciosa* Tenora	Reduced fruit drop, increased fruit set, fruit size and fruit yield	[19,39]
*Manihot utilissima* Pohl.	Increased plant height and leaf number	[82]
*Helianthus annuus* L.	Increased germination %, seedling emergence and survival, plant height, dry shoot mass, dry root mass and yield	[46]
*Basella alba* cv. Red Malabar	Increased plant growth and yield	[81]
*Cucurbita pepo* L.	Produced better growth and seed yield	[83]
*Capsicum annuum* L.	Increased seed germination %, germination index, germination velocity, plant growth, fruit yield and fruit nutrient concentration	[84]
*Foeniculum vulgare* Mill.	Improved vegetative growth and yield characteristics	[24]
*Pelargonium graveolens* L.	Increased plant height, branch number, leaf area, overall plant biomass, volatile oil content and geraniol and citronellol	[6]
*Ocimum basilicum* L.	Improved growth and yield	[54]

## Data Availability

Not applicable.

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
