# Peer review of "Assessing the Usefulness of Moringa oleifera Leaf Extract as a Biostimulant to Supplement Synthetic Fertilizers: A Review"

_plants, 2022, doi:10.3390/plants11172214_

Round 1
Reviewer 1 Report
Dear Authors,
The present study reviews the existing studies on the use of M. oleifera leaves extract as a biostimulant for the enhancement of crop growth and productivity. The research subject is interesting and brings scientific important data in the field, as it deals with a subject that is currently of great interest. Some changes of the manuscript should nevertheless be performed in order to improve its quality. Following specific changes should thus be performed:
Minor changes
All scientific names of genera and species should be italic throughout the whole manuscript. All scientific names of species should have paternity each time they appear for the first time in text (please check especially table 4 and section 9).
You cannot refer to the M. oleifera leaves extract as Moringa leaves extract, as Moringa is a genus ad you deal with a specific species. Please change to M. oleifera leaves extract.
Line 20: please clarify “numerous studies”
Major changes
Abstract: Please specify what do you mean by “the liquid extracted by…” and please be clearer about which type of extract do you refer to. Moreover, the abstract should follow the structure of the manuscript. In the present form, it does not present any materials and methods, results and discussions and conclusions.
Regarding the structure of the manuscript, I recommend following the usual structure of a research article, with Introduction, Materials and Methods, Results and/or Discussions, Conclusions. The actually existing sections may be subsections of the Results and/or Discussions section(s).
Introduction and following sections: Moringa is a genus, not a species. Please refer to the chosen species or begin introducing it by an appropriate introduction of the genus (e.g. line 62, 91, 125, 219) and use italics every time. Please be specific here also about which type of extract does MLE refer to. It is very late to only mention them in section 5-7. It is very vague to only mention “extract”.
Introduction : I noticed that you mention no similar studies were performed in scientific literature. In this case, please add some explanations on what your study brings in novelty. The purpose of the study needs to be rephrased to become clearer (you have specific purposes, treated especially in section 9), which are not found in the explanation of purpose. Please be clearer about the rationale for choosing this species.
3rd section: Please be more specific about the phytochemical composition of the species. It is not clear if the compounds cited in tables are the only ones that are found (no other phytochemical constituents such as flavonoids?). It is also not clear in which way the cited nutrients can be brought to a supposed diet with these fresh/powdered leaves are consumed.
Section 9.1 specifies “studied” cereals and I do not understand the sens of this word here. Maybe “specific” or “selected”?
General remarks: It is not clear whether the biostimulant properties for the enhancement of crop growth and productivity of the M. oleifera leaves extracts are due to the biological activities of the species (e.g. antibacterial ones) or to the nutritional ones (you only treat the use of these extracts in plants with nutritional value). You should emphasize novelty and originality of the present study once again in the specific part of your manuscript.
All these suggested changes should be performed in order to bring further improvements to the manuscript.
Author Response
Dear Reviewer 1
Thank you for your effort and input on our manuscript. We have revised the manuscript based on your recommendation. See below responses
|
Comments from reviewer 1 |
Responses from authors |
|
Reviewer 1 |
|
|
All scientific names of genera and species should be italic throughout the whole manuscript. All scientific names of species should have paternity each time they appear for the first time in text (please check especially table 4 and section 9). |
Authors agree with the reviewer’s comment and changes have been made throughout the manuscript. |
|
You cannot refer to the M. oleifera leaves extract as Moringa leaves extract, as Moringa is a genus ad you deal with a specific species. Please change to M. oleifera leaves extract. |
We don’t agree with the reviewer on this comment. Although Moringa genus has 13 documented species of the Moringaceae family, M. oleifera is widely known and utilised. As such, it has many common names including “moringa”, “drumstick tree”, “miracle tree”, “horseradish tree”, “ben oil tree” and many more (see Price, 2007; Lekgau, 2011; Oyeyinka and Oyeyinka, 2018). |
|
Line 20: please clarify “numerous studies” |
We revised and deleted “numerous studies….” and started the sentence in Line 20 with “It…..” |
|
Abstract: Please specify what do you mean by “the liquid extracted by…” and please be clearer about which type of extract do you refer to. Moreover, the abstract should follow the structure of the manuscript. In the present form, it does not present any materials and methods, results and discussions and conclusions. |
We did not say “the liquid extracted by…” but the “the liquid extracted from…” the type of extracts are well defined within the main text from lines 140-198. Given the word limit of the abstract, it is not feasible to go into greater detail. We could rather just say “the liquid extracted from moringa using different extraction methods…”, however, this will require us to explain such methods as an abstract is supposed to be a standalone We respectfully disagree with the reviewer that the abstract did not follow the structure of the manuscript. The main text of this review does not have materials and methods, results and discussions sections, therefore, it is not possible to have these sections in an abstract. |
|
Regarding the structure of the manuscript, I recommend following the usual structure of a research article, with Introduction, Materials and Methods, Results and/or Discussions, Conclusions. The actually existing sections may be subsections of the Results and/or Discussions section(s). |
The suggested structure by the reviewer is suitable for original papers or meta-analysis papers, and therefore, not suitable for our type, a review paper. |
|
Introduction and following sections: Moringa is a genus, not a species. Please refer to the chosen species or begin introducing it by an appropriate introduction of the genus (e.g. line 62, 91, 125, 219) and use italics every time. Please be specific here also about which type of extract does MLE refer to. It is very late to only mention them in section 5-7. It is very vague to only mention “extract”. |
We agree with the reviewer that Moringa is a genus, not a species. However, in the context of our paper and many published papers, Moringa oleifera is mostly referred to/called by its many common names, of which one is as “moringa”. Moringa is, in our paper, used as a common name for M. oleifera (see Price, 2007; Lekgau, 2011; Oyeyinka and Oyeyinka, 2018 and many more authors that used moringa as a common name of M. oleifera). We agree with the reviewer on mentioning the types of extract of MLE in the introduction section. As such, in Line 66-67 we added “There are various types of MLEs, including aqueous extracts, pressurized hot water extracts and solvents (see section 4 for detailed description)”. |
|
Introduction: I noticed that you mention no similar studies were performed in scientific literature. In this case, please add some explanations on what your study brings in novelty. The purpose of the study needs to be rephrased to become clearer (you have specific purposes, treated especially in section 9), which are not found in the explanation of purpose.
Please be clearer about the rationale for choosing this species. |
Indeed, to the best of our knowledge, no review of this nature has been carried out previously. In Line 74-78, we indicated the purpose of this review which was to gather information on MLE as a natural biostimulant to improve growth, yield and quality of crops under standard and stressful growing conditions. We further mentioned that we will evaluate methods of MLE preparation and preservation of moringa leaves. Further, we highlighted the potential benefits of introducing MLE, particularly to smallholder farmers, as a means of phytonutrient application, while also identifying knowledge gaps that could be explored. The rationale for choosing this species is well outlined in Lines 52-65 and 110-114 in the main text of the paper. |
|
3rd section: Please be more specific about the phytochemical composition of the species. It is not clear if the compounds cited in tables are the only ones that are found (no other phytochemical constituents such as flavonoids?). |
The phytochemical composition presented in section 3 are the commonly detected ones by various researchers and most of these phytochemicals contribute directly to making moringa a potential growth enhancer. Yes, there are some other compounds that the moringa species contain, but they are found in very low amounts and are not of interest to this review. Moringa leaf powder is sold as a commercial product to enhance intake of certain nutrients believed to be beneficial to human health. When used as an extract to improve plant growth and development, MLP can supply certain nutrients essential to plant growth and development. |
|
Section 9.1 specifies “studied” cereals and I do not understand the sens of this word here. Maybe “specific” or “selected”? |
We are referring to all studied cereals, MLEs were tested on. If we use selected or specific species, it will imply that we excluded other types of cereals, which is not the case. |
|
General remarks: It is not clear whether the biostimulant properties for the enhancement of crop growth and productivity of the M. oleifera leaves extracts are due to the biological activities of the species (e.g. antibacterial ones) or to the nutritional ones (you only treat the use of these extracts in plants with nutritional value). You should emphasize novelty and originality of the present study once again in the specific part of your manuscript. |
It is, indeed, not clear, if the biostimulant activity (specifically the cytokinins found in MLE) or simply the added nutrients. As the amount of nutrients is not as much as in a common fertiliser treatment, it is likely the combination of both, added nutrients plus added plant hormones, that results in the ‘moringa effect’. We did not treat the use of MLEs in plants with nutritional value; we only reviewed studies that reported the application of MLE on numerous crops. In those studies, the plants treated with MLE had enhanced growth and productivity when compared to untreated control; and this is clearly described within the manuscript (Line 225- 429). Generally, review papers do not emphasize the novelty and originality of the study. We, therefore, disagree with the reviewer’s suggestion. It is, however, a critical, constructive analysis of the literature in a specific field through summary, classification, analysis, and comparison. Authors rely on previously published literature or data. Review articles are meant to organize and evaluate literature, as well as identify patterns and trends in the literature. Finally, such papers synthesize literature and identify research gaps and recommend new research areas. |
Reviewer 2 Report
This paper reviews the potential of Moringa oleifera leaf extracts as a biostimulant to supplement synthetic fertilisers. The authors pointed out that the leaves of this plant are rich in nutrients and plant hormones at the same time. They also pointed out that both aqueous and methanol extracts of the leaf are effective as biostimulators, but they did not analyse and explain which method of extraction would be the most effective. From the list of references it can be seen that there are a lo of papers dealing with the extraction. My suggestion is to summarize the information provided in the sections 4-7 in a table, and to critically asses them in the text.
Apart from that, it is known that Moringa oleifera, fast-growing, drought-resistant tree native to the Indian subcontinent is widely cultivated for its young seed pods and leaves, used as vegetables and for traditional herbal medicines.
My second question is related to the sustainability of the idea of using leaves of M. Oleifera to produce plant hormones when they are already used as a source of nutrients. Could the water produced during the preparation of food from the leaves of Moringa oleifera be used instead of leaf extracts? What are the key molecules that need to be contained in the extract? These issues need to be addressed especially if this idea is to be introduced to smallholder farmers.
Author Response
Dear Reviewer 1
Thank you for your effort and input on our manuscript. We have revised the manuscript based on your recommendation. See below responses
We thank you
|
Comments from reviewer 2 |
Responses from authors |
|
Reviewer 2 |
|
|
The authors pointed out that the leaves of this plant are rich in nutrients and plant hormones at the same time. They also pointed out that both aqueous and methanol extracts of the leaf are effective as biostimulators, but they did not analyse and explain which method of extraction would be the most effective. |
In section 4, the methods and the preparation of MLE are extensively described. We mention that, even though chemical-based solvents are commonly used, the most effective approach in the extraction of bioactive compounds from moringa leaves is aqueous extraction. This approach is also gaining popularity, as it is environmentally friendly and can be used by both small-scale and commercial farmers. Even though the use of water as a solvent does not result in a high yield of certain bioactive compounds present in moringa leaves, it still is preferred by most farmers. This section further highlights that pressurized hot water extraction is another effective method employed. Yes, the use of chemical-based solvents has been and is still the commonly used method of extraction, but in the current review, this approach is discouraged. |
|
From the list of references, it can be seen that there are a lot of papers dealing with the extraction. My suggestion is to summarize the information provided in the sections 4-7 in a table, and to critically assess them in the text. |
Yes, we agree that this information can be summarised into a single table, but we believe that, if we follow this approach, important information would not appear in the table, or, the table will become too wordy. The way this information is presented in section 4 is detailed and does not force the reader to go to the literature. |
|
My second question is related to the sustainability of the idea of using leaves of M. oleifera to produce plant hormones when they are already used as a source of nutrients. |
Moringa oleifera is both, a rich source of plant hormones and a source of various nutrients, making it an ideal growth regulator. In terms of sustainability, we believe that the idea is sustainable… |
|
Could the water produced during the preparation of food from the leaves of Moringa oleifera be used instead of leaf extracts? |
We are not sure about that, but we believe it can be used since it is similar to aqueous extraction. However, in this review, the focus is primarily on moringa as an alternative to partially substitute or replace chemical-based fertilizers, but not as a food source. Studies focusing on moringa as a food source can investigate this. It must also be noted that not all countries use Moringa oleifera plant parts as a food source. |
|
What are the key molecules that need to be contained in the extract? These issues need to be addressed especially if this idea is to be introduced to smallholder farmers. |
Moringa plant parts especially the leaves contain growth-promoting features, including, but not limited to natural antioxidants, phenolics, mineral nutrients and phytohormones, particularly the cytokinin zeatin; these compounds are believed to play a significant role in enhancing the growth of crops. It is, however, unfortunate that there is no study that compared the compounds or molecules in various MLEs (aqueous and chemical MLE) |
Reviewer 3 Report
The paper "Assessing the Usefulness of Moringa oleifera Leaf Extract as a Biostimulant to Supplement Synthetic Fertilisers: A Review" aims to harvest the research results as a plant biostimulant of Moringa oleifera leaf extract (MLE). A first look observation is that it requires a few relevant schemes and figures to attract the reader.
Also, the review could be more scientifically relevant if it would present some analytical methods for evaluation the specific biocompounds in MLE, for example as a new chapter.
Structurally, i consider that chapters 5,6,7,8 could be presented as sub-chapters of Chapter 4.
L163-164: The phrase misses the predicate and the topic should be revised.
L167-169: The study that firstly used and reported an extraction method should be mentioned first. Same comment for L200 and as a general comment for the entire article.
L191-192: There are two hyperlinks underneath the words of "bioactive compounds" and "moringa leaf", also written with different font, please revise.
In Chapter 12, particular biocompounds and the extraction methods providing the highest yields could be emphasized to increase the interest and impact.
Author Response
Dear Reviewer 3
Thank you for your effort and input on our manuscript. We have revised the manuscript based on your recommendation. See below responses
We thank you
|
Comments from reviewer 3 |
Responses from authors |
|
Reviewer 3 |
|
|
…A Review" aims to harvest the research results as a plant biostimulant of Moringa oleifera leaf extract (MLE). A first look observation is that it requires a few relevant schemes and figures to attract the reader. |
We are grateful for this comment, and we believe the use of schemes and figures would have made our review more persuasive. However, the inclusion of figures and schemes may not represent the core message we aimed to convey to the readers and there is no relevant scheme or figure that we can put in this review. Therefore, we respectively disagree with the reviewer. |
|
Also, the review could be more scientifically relevant if it would present some analytical methods for evaluation the specific biocompounds in MLE, for example as a new chapter. |
The idea is good however, information on analytical methods of leaf extraction is extremely limited, and definitely not sufficient to include in a review, we are, however, currently conducting studies to characterize various MLEs obtained using various solvents. |
|
Structurally, i consider that chapters 5,6,7,8 could be presented as sub-chapters of Chapter 4. |
The authors agree with this suggestion and changes has been made. Below is the new order; Section 5 = 4.1; section 6 = 4.2 and section 7 = 4.3 |
|
L163-164: The phrase misses the predicate and the topic should be revised.
|
This statement has been overhaul for clarity. It now reads Although water as a solvent does not result in a high yield, generally, the aqueous extraction of moringa has gained popularity over the past years amongst farmers, especially subsistence and smallholder farmers, because water is most readily available, environmentally friendly and cheap compared to other solvents such as methanol and alcohol [58,62–64]. Section 4.1, lines 161-164. |
|
L167-169: The study that firstly used and reported an extraction method should be mentioned first. Same comment for L200 and as a general comment for the entire article. |
In the manuscript (section 4.2, line 168- 169) it is clearly stated that “authors used the extraction system using hot water pressure as described by Matshedido et al. [66] with some modifications”. The part is in bold and underlined. Same as in section 5, references 73 and 74 (Foline et al. [73], line 207 and Saini et al. [74], line 211) are the studies that firstly reported the successful preservation methods of particularly moringa leaves. |
|
L191-192: There are two hyperlinks underneath the words of "bioactive compounds" and "moringa leaf", also written with different font, please revise. |
We have attended to the comment (now line 194) |
|
In Chapter 12, particular biocompounds and the extraction methods providing the highest yields could be emphasized to increase the interest and impact. |
In chapter 12 (now revised as chapter 9) lines 452-457, the following phrase has been inserted to address the reviewers comment: “Moringa plant parts, particularly the leaves, contain growth-promoting substances like ascorbic acid, phenolics, natural antioxidants, various mineral nutrients and growth hormones, these bioactive compounds make MLE an ideal plant growth promoter. It must, however, be noted that there are various extraction methods and solvents used to extract the active ingredients in moringa leaves and the aqueous extraction is recommended as it is environmentally friendly and can be used by both subsistence commercial farmers and smallholder farmers in marginal communities”. |
Round 2
Reviewer 1 Report
Dear Authors,
The present study reviews the existing studies on the use of M. oleifera leaves extract as a biostimulant for the enhancement of crop growth and productivity. The authors performed some of the suggested chnges after the first round of review. Other observations were adequately explained. However, some changes of the manuscript should still be performed in order to improve its quality:
Minor changes
If you want to maintain the popular name of the species as “moringa”, you should check it is not italic, as the italic refers to the genus. However, M. oleifera leaves extract is more scientific and therefore more appropriate.
Major changes
Regarding the structure of the manuscript, I still recommend following the usual structure of a research article, with Introduction, Materials and Methods, Results and/or Discussions, Conclusions. In fact, any scientific article should be structured like this, either if it contains practical experiments, or it is a theoretical one. Review papers may have this structure too.
Introduction : The novelty is still not clear. Please specify what you bring in novelty. The purpose of the study still needs to be rephrased to become clearer (you have specific purposes, treated especially in section 9), which are not found in the explanation of purpose.
All these suggested changes should be performed in order to bring further improvements to the manuscript.
Author Response
Dear reviewer
We thank you for your time and contribution to our manuscript. We have provided clarification below.
|
Comments from reviewer 1 |
Response from the authors |
|
|
|
|
If you want to maintain the popular name of the species as “moringa”, you should check it is not italic, as the italic refers to the genus. However, M. oleifera leaves extract is more scientific and therefore more appropriate |
We agree that M. oleifera leaves (leaf not leaves) extract is more scientific and therefore more appropriate, however, we want to maintain the popular name of the species as ‘moringa’ and we have checked that it is not in italics. |
|
|
|
|
Regarding the structure of the manuscript, I still recommend following the usual structure of a research article, with Introduction, Materials and Methods, Results and/or Discussions, Conclusions. In fact, any scientific article should be structured like this, either if it contains practical experiments, or it is a theoretical one. Review papers may have this structure too. |
We agree that some of the review articles follow the usual structure of a research article, with Introduction, Materials and Methods, Results and/or Discussions, and Conclusions, however, our review paper is a critical review, not a systematic or meta-analysis. Our review paper presents an introduction, followed by a discussion of certain aspects of moringa that are of interest in the present review article. Since this is a review of literature not an experimental chapter, no materials and methods, as well as results presented. Lastly, the conclusion is presented.
Most of the recently published review articles in Plants follow the same writing style as in our review article.
Examples of the review articles (presented in the same ways as our review) recently published in Plants are as follows:
Integrated Approach in Genomic Selection to Accelerate Genetic Gain in Sugarcane [(Plants 2022, 11(16), 2139; https://doi.org/10.3390/plants11162139 (registering DOI)]
Description of an Arabica Coffee Ideotype for Agroforestry Cropping Systems: A Guideline for Breeding More Resilient New Varieties [Plants 2022, 11(16), 2133; https://doi.org/10.3390/plants11162133]
Research Progress on the Leaf Morphology, Fruit Development and Plant Architecture of the Cucumber [Plants 2022, 11(16), 2128; https://doi.org/10.3390/plants11162128]
Chilling Tolerance in Maize: Insights into Advances—Toward Physio-Biochemical Responses’ and QTL/Genes’ Identification [Plants 2022, 11(16), 2082; https://doi.org/10.3390/plants11162082] Management and Utilization of Plant Genetic Resources for a Sustainable Agriculture [Plants 2022, 11(15), 2038; https://doi.org/10.3390/plants11152038]
|
|
|
|
|
Introduction: The novelty is still not clear. Please specify what you bring in novelty. The purpose of the study still needs to be rephrased to become clearer (you have specific purposes, treated especially in section 9), which are not found in the explanation of purpose. |
To our understanding, a review article doesn’t necessarily have novelty, but it mostly gathers information, organizes and evaluates literature and finally exposes research gaps and recommends new research to be conducted to fill the exposed gaps. We have, however, revised the purpose or aim of the study (lines 83 -87). Please let us know if you want us to add more.
“We evaluated various methods of extracting bioactive compounds on moringa leaves, highlighting the economically friendly method that can be used by both commercial and smallholder farmers. Furthermore, the preservation methods of moringa leaves were assessed. Finally, we highlighted the potential benefits of introducing it,…”
|
|
|
|
Reviewer 2 Report
The paper can now be accepted
Author Response
We thank you for your valuable input